# Mutagenesis of the Peptide Inhibitor of ASIC3 Channel Introduces Binding to Thumb Domain of ASIC1a but Reduces Analgesic Activity

**DOI:** 10.3390/md22090382

**Published:** 2024-08-24

**Authors:** Timur A. Khasanov, Ekaterina E. Maleeva, Sergey G. Koshelev, Victor A. Palikov, Yulia A. Palikova, Igor A. Dyachenko, Sergey A. Kozlov, Yaroslav A. Andreev, Dmitry I. Osmakov

**Affiliations:** 1Shemyakin–Ovchinnikov Institute of Bioorganic Chemistry, Russian Academy of Sciences, Ul. Miklukho-Maklaya 16/10, 117997 Moscow, Russia; hasanov.ta@phystech.edu (T.A.K.); katerina@1ns.ru (E.E.M.); sknew@yandex.ru (S.G.K.); serg@ibch.ru (S.A.K.); osmadim@gmail.com (D.I.O.); 2Moscow Center for Advanced Studies, Kulakova Str. 20, 123592 Moscow, Russia; 3Branch of the Shemyakin-Ovchinnikov Institute of Bioorganic Chemistry, Russian Academy of Sciences, 6 Nauki Avenue, 142290 Pushchino, Russia; vpalikov@bibch.ru (V.A.P.); yuliyapalikova@bibch.ru (Y.A.P.); dyachenko@bibch.ru (I.A.D.)

**Keywords:** acid-sensing ion channel, peptide ligand, mutagenesis, molecular docking, pain models, analgesic effect

## Abstract

Acid-sensing ion channels (ASICs), which act as proton-gating sodium channels, have garnered attention as pharmacological targets. ASIC1a isoform, notably prevalent in the central nervous system, plays an important role in synaptic plasticity, anxiety, neurodegeneration, etc. In the peripheral nervous system, ASIC1a shares prominence with ASIC3, the latter well established for its involvement in pain signaling, mechanical sensitivity, and inflammatory hyperalgesia. However, the precise contributions of ASIC1a in peripheral functions necessitate thorough investigation. To dissect the specific roles of ASICs, peptide ligands capable of modulating these channels serve as indispensable tools. Employing molecular modeling, we designed the peptide targeting ASIC1a channel from the sea anemone peptide Ugr9-1, originally targeting ASIC3. This peptide (A23K) retained an inhibitory effect on ASIC3 (IC_50_ 9.39 µM) and exhibited an additional inhibitory effect on ASIC1a (IC_50_ 6.72 µM) in electrophysiological experiments. A crucial interaction between the Lys23 residue of the A23K peptide and the Asp355 residue in the thumb domain of the ASIC1a channel predicted by molecular modeling was confirmed by site-directed mutagenesis of the channel. However, A23K peptide revealed a significant decrease in or loss of analgesic properties when compared to the wild-type Ugr9-1. In summary, using A23K, we show that negative modulation of the ASIC1a channel in the peripheral nervous system can compromise the efficacy of an analgesic drug. These results provide a compelling illustration of the complex balance required when developing peripheral pain treatments targeting ASICs.

## 1. Introduction

Acid-sensing ion channels (ASICs) are members of the superfamily known as the degenerin/epithelial Na^+^-channels [1]. In mammals, six isoforms (ASIC1a, ASIC1b, ASIC2a, ASIC2b, ASIC3, and ASIC4) form homo- and heterotrimeric channels, exhibiting variances in both biophysical properties and involvement in physiological and pathological processes [2]. ASIC3 channels are extensively distributed in the peripheral nervous system (PNS) and are implicated in nociception and inflammation processes [3,4]. ASIC1a channels are prevalent in both the PNS and CNS, contributing to synaptic plasticity [5], fear perception [6], epilepsy [7], oncology [8], and various other diseases [9]. Nonetheless, the precise mechanism of ASIC1a involvement in PNS processes remains unclear.

In 2018, the structure of full-length chicken ASIC1a (cASIC1a) was elucidated using X-ray crystallography and cryoelectron microscopy (cryo-EM) [10]. The channel comprises short intracellular N- and C-terminal regions, two transmembrane domains (TMs), and an extracellular domain (ECD). The ECD consists of the finger, thumb, knuckle, palm, β-ball, and wrist domains. Notably, four pairs of spatially proximal amino acid residues, including Asp238–Asp350 and Glu239–Asp346 in the finger and thumb domains of one subunit and Glu220–Asp408 and Glu80–Glu417 in the palm domain of the other subunit, collectively form the “acidic pocket”, crucial for proton stimulus perception [11].

Selective ligands targeting specific ASIC isoforms are invaluable for structural and functional investigations. Peptide ligands sourced from various animal venoms have emerged as such tools. For instance, Psalmotoxin-1 (PcTx1), a peptide toxin from tarantula venom, with the inhibitor cystine knot motif, selectively modulates ASIC1a and ASIC1a-containing channels by stabilizing the desensitized state [11,12,13]. Conversely, Mambalgin-1 (Mamb-1), a three-finger peptide from snake venom, also negatively modulates ASIC1-containing channels but by stabilizing the closed state [14,15]. Both PcTx1 and Mamb-1 demonstrate analgesic effects, elucidating the significance of ASIC1a channels in the CNS in pain perception [16,17].

The 29-amino acid peptide Ugr9a-1, isolated from the venom of the northern sea anemone *Urticina grebelnyi*, represents a structural class of toxins termed “boundless β-hairpin”. Ugr9-1 selectively modulates ASIC3 channels and exhibits promising results in mitigating acid-induced and inflammatory pain in mouse models [18,19,20].

In this study, we aimed to elucidate why Ugr9-1 lacks activity towards ASIC1a channels. Through a combination of bioinformatics analysis, molecular modeling, and functional mutagenesis, we discovered that substitution of a lysine residue at position 23 in the peptide (A23K) enables inhibition of ASIC1a alongside ASIC3 channels. Furthermore, investigation of the mutant ASIC1a channel revealed a specific site within the thumb domain responsible for interaction with A23K. However, analysis of analgesic activity in vivo revealed that the mutant peptide A23K, active on both ASIC1a and ASIC3 channels, was inferior to the wild-type peptide, which solely targets ASIC3.

## 2. Results

### 2.1. Changing the Selectivity of the Ugr9a-1 Peptide Based on Comparison with ASIC1a Channel Inhibitors

This study aimed to elucidate the basis of the specificity of the Ugr9a-1 peptide (Ugr9-1) that selectively inhibits the ASIC3 channel without affecting ASIC1a channel activity. Initially, we conducted an analysis of the amino acid composition of established ASIC1a channel inhibitors, namely PcTx1 (known for its selective inhibition) and Mamb-1. This analysis revealed a critical disparity: the absence of lysine residues in the sequence and the overall negative charge of Ugr9-1 (Figure 1). Previous research highlighted the significance of lysine residues in PcTx1 and Mamb-1 for ASIC1a channel inhibition; substitutions with alanine markedly diminished inhibitory efficacy by 1.7–97.7 times [21,22,23].

We hypothesized that the presence of lysine residues in various regions of Ugr9-1—whether in the N-terminal (D4K), the protruding loop (S16K), or forming a basic aromatic cluster in the C-terminal (V20K, A23K)—might confer the ability to modulate ASIC1a channel activity (Figure 1).

To validate the hypothesis, we used the model structure of the rat ASIC1a channel (rASIC1a). Subsequently, this model was utilized for molecular docking of Ugr9-1 (PDB code 2LZO), both the wild-type Ugr9-1 and its mutants (D4K, S16K, V20K, and A23K). The findings revealed that Ugr9-1 (Figure 2A) along with its D4K and V20K mutants apparently bind to rASIC1a within the transmembrane domain or intracellular segment (Figure A1). Conversely, docking models for the S16K and A23K mutants indicated interactions with residues D355 and E357 of the “thumb” domain on the channel. Specifically, for the S16K mutant, bond lengths between the K16 residue on the peptide and the D355 and E357 residues on the channel measured 11.4 and 8.0 Å, respectively, while for the A23K mutant, bond lengths between K23 on the peptide and D355 and E357 on the channel were 7.5 and 10.8 Å, respectively (Figure 2B,C). The calculated energy parameters indicate a significant difference in the interaction scores between the peptides and the channel. For the peptides S16K and A23K, the average interaction score is less than −17 (Table 1). In contrast, the scores for the models of other peptide molecules only reach up to −14 (Table 1 and Table A1).

Thus, molecular modeling confirmed the inactivity of Ugr9-1 against the ASIC1a isoform and allowed us to speculate that lysine substitutions at positions 16 and 23 may confer the ability of the peptides to interact with rASIC1a, whereas substitutions at positions 4 and 20 do not.

### 2.2. Mutations Alter the Activity and Selectivity of the Ugr9-1 Peptide

Based on molecular docking results, we synthesized recombinant analogues of the S16K and A23K mutants. Expression of the mutant genes was conducted in *E. coli* cells, followed by purification of the final products using RP-HPLC (Figure A2). The yield of mutant peptides was 5 mg/L of culture, matching that of Ugr9-1.

To evaluate peptide activity, we performed electrophysiological testing using the two-electrode voltage clamp technique on *X. laevis* oocytes expressing rat ASIC channels. Consistent with prior findings [18], Ugr9-1 exhibited no activity towards rASIC1a. However, both the S16K and A23K mutants demonstrated the ability to inhibit this channel (Figure 3A). The inhibitory effect was concentration-dependent, with calculated half-maximal inhibitory concentration (IC_50_) values of 22.69 ± 0.42 µM (Hill slope (n_H_) of 2.86 ± 0.09) for S16K and 6.72 ± 0.38 µM (n_H_ value of 2.84 ± 0.31) for A23K (Figure 3B).

Furthermore, examination of the interaction between peptides and the rat ASIC3 channel (rASIC3) revealed that the S16K and A23K mutants, akin to the original Ugr9-1, inhibited currents through the ASIC3 channel (Figure 3C). The IC_50_ values were 35.77 ± 4.03 µM (n_H_ 2.16 ± 0.24) for S16K and 9.39 ± 0.07 µM (n_H_ 3.58 ± 0.08) for A23K (Figure 3D). For comparison, Ugr9a-1 inhibited rASIC3 with an IC_50_ value of 14.16 ± 0.96 µM (n_H_ of 2.33 ± 0.17). Notably, S16K and A23K did not exhibit activity on the rat ASIC1b channel (Figure A3).

In line with molecular docking predictions, the mutant peptides S16K and A23K demonstrated the ability to modulate the ASIC1a isoform while retaining activity towards the ASIC3 isoform. Moreover, the structural interaction of the mutants with the channel, weaker for S16K and stronger for A23K in silico (Figure 2B,C), was validated in vitro through their inhibitory effect on channel function.

The activity of the A23K mutant peptide was also evaluated on rat ASIC1a/3 heteromeric channels expressed in *X. laevis* oocytes. The peptide inhibited these channels with IC_50_ of 4.2 ± 0.6 µM and nH 1.6 ± 0.2 (Figure 4). Thus, the inhibitory effect of A23K on heteromeric channels was approximately 1.6-fold and 2.2-fold more effective compared to its effect on homomeric ASIC1a and ASIC3 channels, respectively. The extended wash procedure required to restore ASIC1a/3 function could be additional evidence of high affinity of A23K to heteromeric channels (Figure 4A).

### 2.3. Residue D355 of the Thumb Domain Is Critical for Ligand Binding to the ASIC1a Channel

Molecular modeling elucidated the involvement of residues D355 and E357 of the thumb domain in the binding of mutant peptides S16K and A23K to rASIC1a. To validate this, we made alanine mutants D355A and E357A of the channel, and both mutants exhibited retained functional activity. For the wild-type channel, the half-maximal pH values of steady-state desensitization (pH_SSD50_) and activation (pH_act50_) were calculated as 7.20 ± 0.02 (n_H_ 11.9 ± 3.6) and 6.69 ± 0.02 (n_H_ 3.56 ± 0.59), respectively. Comparatively, mutant channels displayed pH_SSD50_ values of 7.13 ± 0.01 (n_H_ 8.02 ± 0.85) for D355A and 7.17 ± 0.01 (n_H_ 7.4 ± 0.6) for E357A; pH_act50_ values were 6.39 ± 0.02 (n_H_ 3.14 ± 0.23) for D355A and 6.61 ± 0.03 (n_H_ 3.84 ± 0.68) for E357A (Figure 5A,B).

The A23K mutant peptide showed the most potent effect on rASIC1a. At a concentration of 10 µM, A23K demonstrated an inhibitory effect of 74.9 ± 6.5% on the wild-type channel (Figure 5C,F). Under the same conditions, the inhibitory effect of A23K on the E357A mutant channel was 47.1 ± 7.7% (Figure 5E,F). However, the most notable observation was with the D355A mutant channel, where A23K exhibited no significant effect (Figure 5D,F). These results not only validate the molecular docking predictions regarding the binding site but also pinpoint a crucial residue on the channel responsible for the ligand binding.

### 2.4. Loss of Selectivity to ASIC3 Channel Leads to the Impaired Analgesic Effect

The original peptide Ugr9-1 is a selective negative modulator of ASIC3 channels and produces a pronounced analgesic effect in both the acetic acid-induced writhing (AAW pain model) and complete Freund’s adjuvant (CFA)-induced hyperalgesia [18,19,20]. The A23K peptide exhibits similar activity on the ASIC3 isoform as Ugr9-1 but also inhibits the ASIC1a isoform. Therefore, it was of interest to investigate how the additional targeting of the ASIC1a channel might alter the analgesic properties of the compound.

In both the AAW and CFA-induced hyperalgesia, Ugr9-1 and A23K peptides were administered intramuscularly one hour before the experiments at doses of 0.002, 0.02, and 0.2 mg/kg. In the AAW test, analgesic efficacy was evaluated by assessing the reduction in the number of involuntary writhing episodes induced by intraperitoneal injection of a 0.6% acetic acid solution. The original Ugr9-1 significantly decreased the number of writhes at all three doses, with the maximum effect (30% inhibition) observed at a dose of 0.02 mg/kg. In contrast, A23K exhibited a significant reduction in writhing episodes only at a dose of 0.02 mg/kg, albeit 1.5 times less effectively than Ugr9-1 (Figure 6A).

In the CFA-induced hyperalgesia test, where CFA administration induces excess sensitivity to mechanical and thermal stimuli, the analgesic effects of peptides were assessed using von Frey filaments of varying forces (von Frey test) and a hot plate apparatus (hot plate test). In the von Frey test, administration of Ugr9-1 led to a significant increase in paw withdrawal threshold (PWT) at all doses, with a maximum effect (270% increase) observed at 0.02 mg/kg. Conversely, A23K significantly increased PWT only at 0.02 mg/kg, more than three times less effectively than Ugr9-1 (Figure 6B). In the hot plate test (53 °C), Ugr9-1 significantly prolonged paw withdrawal time at all doses, with the maximum effect (106% increase) observed at 0.02 mg/kg, while A23K showed effects only at doses of 0.02 and 0.2 mg/kg, two times less effective than Ugr9-1 (Figure 6C).

Overall, in all pain tests conducted, the A23K peptide produced a significantly less pronounced analgesic effect compared to the wild-type peptide. This suggests that negative modulation of ASIC1a channels in the peripheral system may decrease the effect of compounds affecting the ASIC3 channel.

## 3. Discussion

Peptide Ugr9-1 was originally identified as a negative modulator of human ASIC3 channels. We demonstrated for the first time the ability of Ugr9-1to effectively inhibit both human and rat ASIC3 channels with similar potency (IC_50_ of 10 µM for human [18] and 14 µM for rat orthologue). Combining comparative analysis, molecular modeling techniques, and structural and functional studies, we generated two novel peptide variants derived from Ugr9-1 capable of modulating both ASIC3 and ASIC1a channels. Previously, peptides with ASIC3 inhibitory properties were made from the inactive homologue of Ugr9-1 [19], highlighting the significance of the Phe9 and His12 residues. In this research, we aimed to engineer a ligand targeting ASIC1a. Through a comparative analysis with well-established peptide ligands PcTx1 and Mamb-1, we strategically introduced lysine residues into the Ugr9-1 molecule at key positions: the N-terminal region (D4K substitution), the top of the β-hairpin loop (S16K substitution), and the C-terminal region (V20K and A23K substitutions), potentially forming a basic aromatic cluster. Molecular docking simulations of these mutant peptides, alongside the original Ugr9-1, with the rat ASIC1a channel in a closed state, revealed that D4K and V20K mutants exhibited preferential interactions with the transmembrane domain, thus were excluded from further consideration. Notably, Ugr9-1 also showed a similar interaction pattern, confirming the lack of activity towards ASIC1a channels.

Models incorporating S16K and A23K revealed compelling intermolecular interactions with residues Asp355 and Glu357 located within the thumb domain of the ASIC1a channel. This domain comprises two α-helices (α4 and α5) linked by a small loop and disulfide bridges, with these helices connected to β-strands (β9 and β10) in the adjacent palm domain [11]. Activation of the channel triggers a conformational shift in the thumb domain, acting as a barrier that restrains the movement of the lower part of the palm domain and the pore under continuous exposure to an acidic milieu, thereby exerting a pivotal role in channel function [10,24]. Prior studies have elucidated that peptide ligands of ASIC1a channels establish contacts with amino acid residues located in the thumb domain. For instance, PcTx1 binds to the ASIC1a channel through interactions involving Trp7 and Trp24 peptide residues and the Phe350 channel residue, facilitating convergence of a hairpin rich in arginine with the acidic pocket region [25]. This positioning induces conformational alterations in the acidic pocket (transitioning to a desensitized/open state), rendering it less receptive to protons [26,27]. Similarly, Mamb-1 engages in interactions within the thumb domain, albeit through a distinct mechanism. Binding of Mamb-1 induces a closed-like state of the channel by locking the hinge between α4 and α5 helices in the thumb domain. Mamb-1 interacts with four key residues of the channel, including Tyr358, Asn320, Tyr316, and Phe350. In particular, Lys8 of Mamb-1 interacts with Tyr358 in rASIC1a, which is closely connected to the α4 helix through its contact with Tyr316, and Leu32 in Mamb-1 also forms a contact with Phe350 in the α5 helix [22,23].

Electrophysiological experimentation confirmed the activity of recombinant S16K and A23K mutants against rat ASIC1a channels. A23K was found to be over three times more active than S16K. This fact was in good agreement with molecular docking results that predicted tighter interactions of the lysine residue at position 23 with Asp355 and Glu357 residues of the channel. Notably, site-directed mutagenesis of the channel involving D355A, E357A (for Mamb-1), and E357N (for Mamb-2) did not substantially alter the inhibitory potential of both mambalgins [23,28]. To confirm the critical role of these residues in channel interaction with A23K, we generated mutant channels with substitutions at positions 355 or 357 to alanine. Channels with such mutations retained their functions the same as was reported before [23]. However, while these substitutions had minimal impact on Mamb-1’s action [23], we revealed that the inhibitory effect of A23K was entirely abolished for the D355A mutant channel.

ASIC1a and ASIC3 channels are important players in pain perception within the peripheral nervous system. While the significant involvement of ASIC3 channels in acid-induced, chronic neuropathic and inflammatory pain has been extensively reported, the role of ASIC1a remains less clear [29]. Direct activation of ASIC1a channels, such as with the MitTx toxin, elicits acute pain reactions upon injection into mouse paws, an effect absent in mice lacking the ASIC1a channel [30]. Conversely, systemic administration of an ASIC1a inhibitor, like mambalgin, reduces neuropathic and inflammatory pain, with similar effects observed even in mice lacking ASIC1a channels [17]. Therefore, the comparison of the effects of Ugr9-1, an ASIC3 inhibitor, with its mutant A23K, which inhibits both ASIC1a and ASIC3 channels, in pain models in vivo becomes particularly intriguing. Prior in vivo studies have shown that Ugr9-1 injection attenuates acid-induced and CFA-induced inflammatory pain in mice across a range of doses [18,20]. In this study, we specifically examined these pain models, assessing parameters including the number of writhing episodes, paw withdrawal threshold, and paw withdrawal time to compare the effects of Ugr9-1 and A23K. Consistent with previous findings, Ugr9-1 demonstrated effective analgesic effects in both pain models. Intriguingly, while the mutant A23K exhibited analgesic effects at certain doses, it demonstrated less potency in all measured parameters than the wild-type peptide activity. A23K demonstrated the most significant effect on thermal hypersensitivity caused by inflammation. Notably, alongside other ASIC1a inhibitors like the low molecular weight alkaloid lindoldhamine or the lignan sevanol, substantial analgesic effects were also noted in this experimental model [31,32]. But, in the context of acid-mediated pain and mechanical hypersensitivity due to inflammation, the analgesic effect of this mutant was compromised, particularly at higher doses. Overall, these findings suggest that the additional activity exhibited by the ligand on ASIC1a channels, beyond the effect on ASIC3, may yield unpredictable results in vivo. This underscores the complex and controversial role of ASIC1a in peripheral pain processes.

## 4. Materials and Methods

### 4.1. Site-Directed Mutagenesis of Peptides

To introduce S16K and A23K substitutions, we designed two pairs of primers. For S16K substitution, the forward primer (F1) sequence was 5′–GAATTAGATCTCATGATTTCCATTGATCCGCCGTGCCGTTTTTGCTATCAT–3′ and the reverse primer for S16K was 5′–TACCACCGTGACGGTTCTGGTAACTGCGTTTACGACAAATACGGTTGCGGT–3′. For A23K substitution, the forward primer sequence was the same as for S16K (designated as “F1”), and the reverse primer for A23K was 5′–GGATTCCTCGAGCTACACCGCGCCGCAGCCATATTTATCATACACGCAATT–3′. These primers were utilized with the plasmid pET32b(+)-Ugr9-1 as a template. Mutant peptides were generated using recombinant PCR strategies as outlined previously [19]. The resulting constructs were validated through sequencing.

### 4.2. Site-Directed Mutagenesis of Rat ASIC1a

To introduce D355A and E357A substitutions, we designed two pairs of primers. For D355A substitution, the forward primer sequence was 5′–TGGAGAAAGCCCAGGAATACTGCG–3′, and the reverse primer (R2) for D355A was 5′–CTAGGAAGTCCAGGGCAG–3′. For E357A substitution, the forward primer sequence was 5′–TGGAGAAAGACCAGGCCTACTGCGTG–3′, and the reverse primer for E357A was the same as for D355A (designated as “R2”). Point mutations were conducted in a pCI-rASIC1a plasmid via PCR utilizing standard protocols with Tersus polymerase (Evrogen, Moscow, Russia). Linear PCR products underwent phosphorylation with PNK and subsequent circularization with T4 ligase. Plasmids containing mutant channels were subjected to sequencing for confirmation of mutagenesis.

### 4.3. Recombinant Peptide Production

The recombinant peptides were expressed as a fusion with thioredoxin and a His-tag in *Escherichia coli* BL-21 as previously described [19]. Transformed cells were cultured in LB medium supplemented with 100 μg/mL ampicillin until reaching an optical density of approximately 0.6–0.8 at 37 °C, with moderate aeration and stirring. Protein synthesis was induced by adding 0.2 mM isopropyl 1-thio-β-d-galactopyranoside. The bacterial culture was subsequently disrupted by ultrasonication. The lysate was then applied to a column packed with HisPur™ Ni-NTA metal affinity resin (Thermo Scientific, Waltham, MA, USA) and eluted according to the manufacturer’s instructions. The fusion protein was cleaved overnight using BrCN. Purification was performed using reverse-phase high-performance liquid chromatography (RP-HPLC) with a linear gradient of acetonitrile in the presence of 0.1% trifluoroacetic acid on a Jupiter C5 column (250 × 10 mm, Phenomenex), followed by further purification on a Luna C18 column (250 × 4.6 mm, Phenomenex). The purity of the recombinant peptide was confirmed using electrospray ionization mass spectrometry.

### 4.4. Isolation of Xenopus laevisoocytes and mRNA Injection

All procedures involving female *X. laevis* adhered to the guidelines outlined in ARRIVE (Animal Research: Reporting of In Vivo Experiments) and the “European Convention for the Protection of Vertebrate Animals Used for Experimental and Other Scientific Purposes” (Strasbourg, 18.III.1986). These protocols were executed following approval by the Institutional Animal Care and Use Committee (IACUC) of the Shemyakin–Ovchinnikov Institute of Bioorganic Chemistry, Russian Academy of Sciences (Protocol Number: 351/2022; date of approval: 24 November 2022).

A small portion of the ovary was isolated from frogs under anesthesia induced by a 0.17% solution of procaine methanesulphonate (MS222) (Sigma), followed by treatment with collagenase (1 mg/mL) at room temperature for 2 h. Oocytes at stages IV and V, devoid of follicular membranes, were transferred to ND-96 medium (containing 96 mM NaCl, 5 mM HEPES, 2 mM KCl, 1.8 mM CaCl_2_, 1 mM MgCl_2_, pH 7.4). mRNA synthesis was performed from pcDNA3.1 vectors (Invitrogen, Carlsbad, CA, USA) containing the rat ASIC3 gene, as well as pCI vectors containing genes for rat ASIC1a, its mutant analogues, and ASIC1b. mRNA (2.5–10 ng) was then injected into the oocytes. For the ASIC1a/3 heteromeric channels, the mRNA encoding the ASIC1a and ASIC3 subunits were injected in a 1:5 ratio, respectively [33]. Subsequently, the oocytes were incubated for 2–3 days at a temperature of 17–19 °C in ND96 medium supplemented with gentamicin and streptomycin (20 µg/mL each).

### 4.5. Electrophysiological Recordings

Whole-cell electrophysiological recordings of ASIC channels were conducted using a two-electrode voltage-clamp setup. Microelectrodes were filled with 3 M KCl. Recordings from oocytes were carried out at a holding potential of −50 mV, with data filtered at 10 Hz and sampled at a rate of 100 Hz using an L780M ADC (LCard, Moscow, Russia). The synchronized delivery of solutions into the chamber and recording of electrophysiological results were carried out using in-house software v.2. Activation of ASIC channels occurred through rapid pH changes in the solution, achieved by replacing the working solution in the chamber with ND96-based solutions at varying pH levels. Specifically, HEPES was substituted with 10 mM MES for pH 5.5–6.5 and MOPS for pH 6.7–7.0.

Fresh peptide solutions were meticulously prepared in ND96 buffer with a pH of 7.43 immediately before experiments, with strict control over the pH levels. Peptides were applied for 30 s at a conditioning pH of 7.43 before acid stimulation of the channel. The interval between applications was maintained at a minimum of 1.5 to 2 min.

### 4.6. Animal Tests

This study adhered strictly to the World Health Organization’s International Guiding Principles for Biomedical Research Involving Animals. Conducted within an Association for Assessment and Accreditation of Laboratory Animal Care International (AAALAC) accredited facility, the research followed the standards outlined in the Guide for Care and Use of Laboratory Animals (8th edition, Institute for Laboratory Research of Animals). Approval for all experiments was obtained from the Institutional Animal Care and Use Committee of the Branch of the Shemyakin–Ovchinnikov Institute of Bioorganic Chemistry of the Russian Academy of Sciences (identification code: 353/2023; approval date: 3 July 2023).

Adult male ICR mice (Animal Breeding Facility Branch of Shemyakin–Ovchinnikov Institute of Bioorganic Chemistry, Russian Academy of Sciences, Pushchino, Russia), weighing 20–25 g, were utilized. Animals were housed under controlled conditions, with a room temperature of 23 ± 2 °C, a 12 h light–dark cycle, and ad libitum access to food and water. Test substances were dissolved in saline. Peptides (Ugr9-1 and A23K at doses of 0.002, 0.02, and 0.2 mg/kg) and saline (for the control group) were administered intramuscularly 60 min prior to testing.

The abdominal constriction test of visceral pain (acetic acid-induced writhing pain model) and hyperalgesia induced by complete Freund’s adjuvant (CFA-induced inflammatory pain model) were performed as previously described [32].

For the von Frey filaments test, mice were positioned in a plastic chamber with access to the inflamed hind paw surface and an electronic von Frey filament (BIO-EVF, Bioseb, Vitrolles, France) was applied with gradually increasing pressure until withdrawal. Data, expressed as the strength of the applied mechanical stimulus from three applications, were averaged, with intervals between exposures of at least 1 min.

Statistical analysis was conducted using analysis of variance (ANOVA) with post hoc Tukey’s test, and data are presented as mean ± standard deviation (S.D.).

### 4.7. Molecular Docking of Peptide Ligands with the Rat ASIC1a Channel

Molecular docking was employed to investigate the interaction between peptide ligands and the rat ASIC1a channel. Mutant analogues of Ugr9a-1 (PDB ID: 2LZO) were constructed using PyMOL v.3.0 (Schrödinger LLC, New York, NY, USA) software. The structural model of rat ASIC1a (UniProt code P55926) was obtained from the AlphaFold2 (ColabFold v1.5.5) [34]. The N-termini (1–40) and C-termini (460–526) of rASIC1a were removed due to the fact that there are no reliable coordinates for these regions. The Rosetta v.3.14 software package was employed for the molecular docking of Ugr9a-1 and its mutants. Structural preparations of Ugr9a-1, its mutant analogues, and the rASIC1a channel involved initial relaxation using the Rosetta Relax program. All five channel–peptide complexes were prepared with PyMOL and saved in .pdb format. Molecular docking was conducted using the RosettaDock-5.0 program [35]. Docking was performed without a lipid bilayer. Subsequently, 1000 docking models were generated for each peptide. The top five models, exhibiting the highest negative ratio of interaction scores to RMSD, were selected (Figure A4), as they indicate proximity to the equilibrium state. These models were then analyzed to assess the interactions between the channel’s amino acid residues and the peptide’s amino acid residues, particularly focusing on the mutated lysine residues.

### 4.8. Data Analysis

Electrophysiological data analysis was conducted using OriginPro 2018 software (OriginLab, Northampton, MA, USA). Dose–response relationships for Ugr9a-1, S16K, and A23K peptides were fitted using a Hill function: F_1_(x) = A − A/(1 + (x/IC_50_)^n_H_), where F_1_(x) represents the amplitude of the current at concentration x of the peptide, A is the maximum amplitude of the current, IC_50_ is the half-maximum inhibitory concentration of the peptide, and n_H_ is the Hill coefficient.

The pH–response dependence of the current for the ASIC1a channel and its mutants was analyzed using the Hill equation F_2_(x) = ((A_1_ − A_2_)/(1 + (x/[H^+^]_50_)^n_H_)) + A_2_, where [H^+^]_50_ denotes the proton concentration at which half-maximal current was reached, A_1_ represents the minimum response value, and A_2_ denotes the maximum response value. The maximum value (I_max_) was calculated from the current amplitude values obtained at each pH for each cell through individual fitting. Subsequently, the data were normalized to the calculated I_max_ value, averaged, and fitted to the equation F_2_(x). The results are presented as mean ± SEM.

## 5. Conclusions

In this study, we employed molecular modeling techniques and structural-functional analysis to engineer mutant analogues of the peptide molecule Ugr9-1 showcasing an inhibitory effect on ASIC1a channels. By mutating various residues to lysine, we enhanced the positive charge of Ugr9-1, forming a basic-hydrophobic cluster conducive to binding ASIC1a channels. Notably, two peptides, A23K and S16K, exhibited significant inhibitory effects on ASIC1a, underscoring the critical role of this cluster in interacting with ASIC channels. We determined a key residue, D355, within the ASIC1a channel responsible for interacting with mutant peptides, highlighting the substantial contribution of the thumb domain in mediating ligand interaction. Despite the efficacy of the A23K peptide in inhibiting ASIC1a and ASIC3 channels in vitro, A23K demonstrated significantly less potency than the parent Ugr9-1 molecule in various pain models, suggesting a nuanced relationship between ASIC modulation and analgesic effects. Our findings demonstrate the intricate pharmacological effects of ASIC modulation in the peripheral nervous system, indicating that negative modulation of ASIC1a may potentially impact the efficacy of drugs targeting ASIC3 channels. To robustly validate this hypothesis, further investigation into the effects of A23K on pain modulation in ASICs knockout mice is warranted. These results emphasize the importance of considering such complexities in the development of analgesic medications.

## Figures and Tables

**Figure 1 marinedrugs-22-00382-f001:**
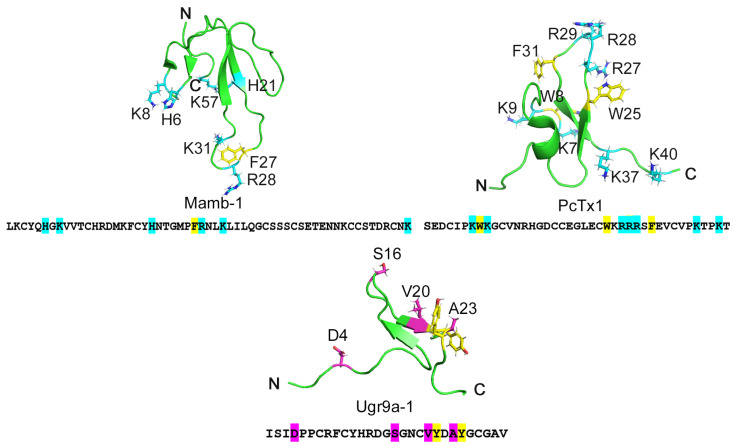
The structures of three peptides: Mamb-1 sourced from snake venom, PcTx1 from tarantula venom, and Ugr9-1 from sea anemone venom. The primary and spatial structures of the peptides inhibiting ASIC channels: Mamb-1, PcTx1, and Ugr9a-1 (PDB codes 1MJU, 2KNI, and 2LZO, respectively). Positively charged residues important for inhibition of the ASIC1a channel are highlighted in blue in Mam-1 and PcTx1. Residues earmarked for substitution in the Ugr9-1 structure are highlighted in pink. Additionally, tyrosine residues, putatively forming a basic-aromatic cluster within the Ugr9-1 structure, are denoted in yellow.

**Figure 2 marinedrugs-22-00382-f002:**
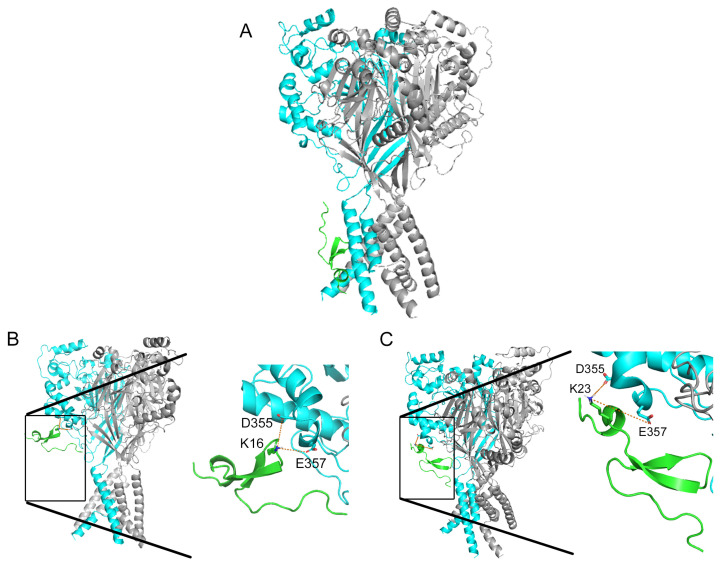
Molecular docking of the wild-type Ugr9-1 (**A**) alongside its mutants S16K (**B**) and A23K (**C**) to the rat ASIC1a channel in closed state. The representation showcases the three channel subunits (cyan and gray levels) and peptides (green) depicted in ribbon form. In (**A**), the Ugr9-1 is observed positioned within the region of transmembrane domains. Conversely, in mutants S16K (**B**) and A23K (**C**), lysine residues at positions 16 and 23, respectively, are presumed to instigate interactions with residues D355 and E357 of the channel. Distances are delineated by orange dotted lines.

**Figure 3 marinedrugs-22-00382-f003:**
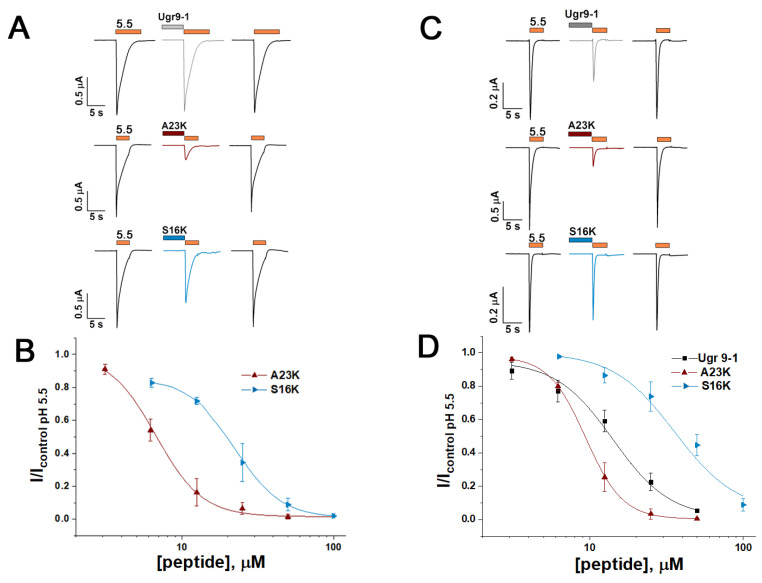
The effect of mutant peptides S16K and A23K on rat ASIC1a and ASIC3 channels. (**A**) Representative current traces demonstrating the effects of 12.5 µM S16K and A23K on ASIC1a as well as the lack of effect observed with 50 µM Ugr9-1 on ASIC1a. (**B**) Dose–response curves for the inhibitory effect of mutant peptides on ASIC1a. Each data point represents data from 5 cells. (**C**) Representative current traces of Ugr9-1, S16K, and A23K at a concentration of 12.5 µM on ASIC3. (**D**) Dose–response curves revealing the inhibitory effect of both Ugr9-1 and mutant peptides on ASIC3. Each point represents data from 5 cells. The depicted currents were induced by a pH 5.5 stimulus following a conditioning pH of 7.4, with peptides applied for 30 s before the pH stimulus. The holding potential (Vh) was −50 mV, and the time interval between applications ranged from 1 to 2 min. Data are presented as mean ± S.E.M.

**Figure 4 marinedrugs-22-00382-f004:**
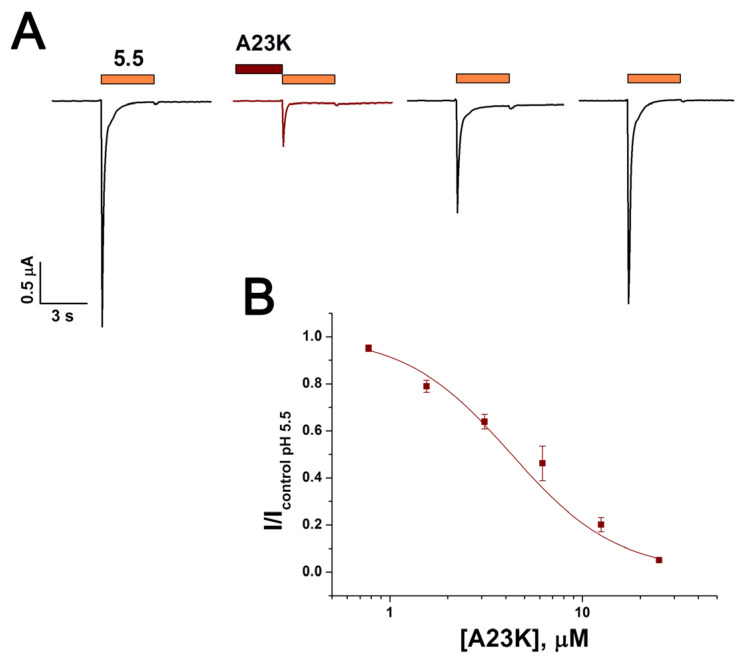
The effect of the mutant peptide A23K on rat heteromeric ASIC1a/3 channels. (**A**) Representative current traces demonstrating the effect of 12.5 µM A23K. The depicted currents were induced by a pH 5.5 stimulus following a conditioning pH of 7.4, with the peptide applied for 30 s before the pH stimulus. (**B**) Dose–response curve for the inhibitory effect of the mutant peptide. Each data point represents data from 6 cells. Data are presented as mean ± S.E.M.

**Figure 5 marinedrugs-22-00382-f005:**
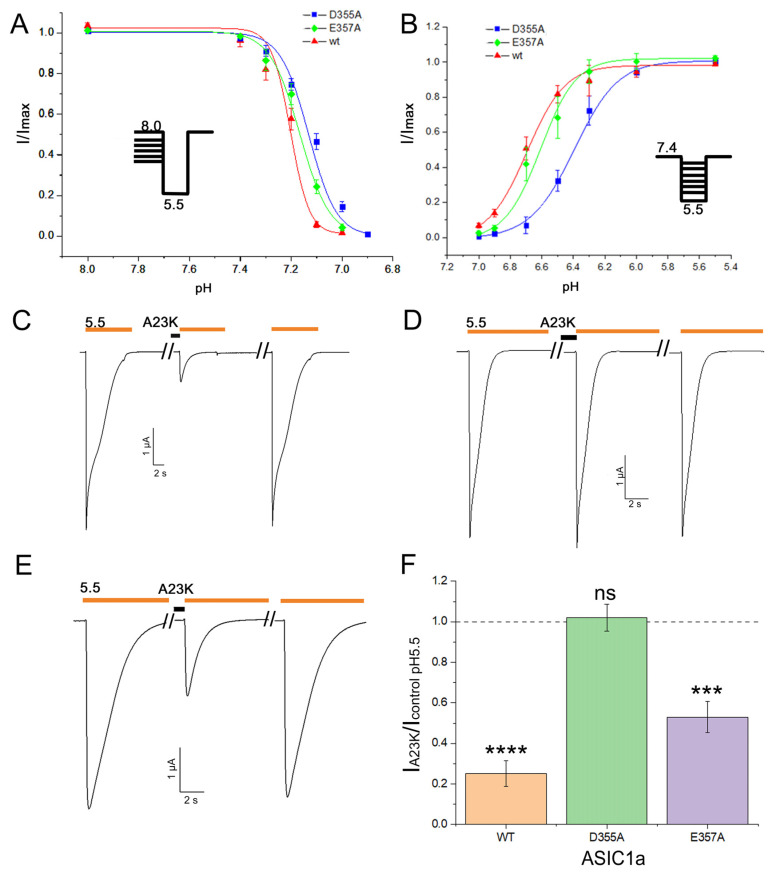
The pivotal role of residue D355 in the ASIC1a channel for the action of the A23K peptide ligand. (**A**,**B**) pH dependence of steady-state desensitization (**A**) and activation (**B**) in mutant channels (D355A and E357A) compared to the wild-type channel (wt). Each data point represents results from 5 cells. (**C**–**E**) Representative current traces illustrating the effects of A23K at a concentration of 10 µM on wt (**C**), D355A (**D**), and E357A (**E**) channels. Currents were evoked by a pH 5.5 stimulus following a conditioning pH of 7.4, with the peptide applied for 30 s preceding the pH stimulus. (**F**) Bar graph quantifying the effects depicted in (**C**,**D**) as a percentage of the corresponding control currents (*n* = 5). A value of 1.0, represented by a dotted line, corresponds to control measurements at pH 5.5 for each form of ASIC1a. The data are expressed as mean ± SEM; *** *p* < 0.001, **** *p* < 0.0001, ns non-significant vs. control, unpaired *t*-test.

**Figure 6 marinedrugs-22-00382-f006:**
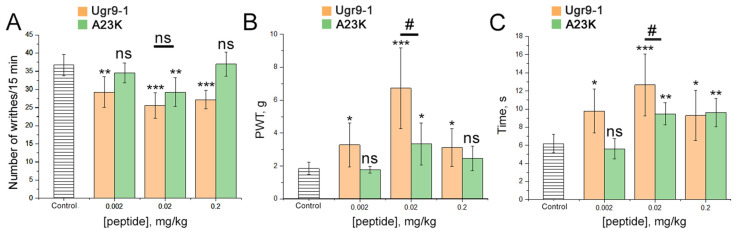
Effects of Ugr9-1 and A23K peptides in pain models. Peptides (Ugr9-1–orange bars, A23K–green bars) were administered intramuscularly one hour before testing. (**A**) The efficacy of peptides in the acetic acid-induced writhing test, evaluated based on the number of writhes following intraperitoneal administration of acetic acid. (**B**,**C**) Peptide effectiveness in hyperalgesia induced by complete Freund’s adjuvant assessed by the paw withdrawal threshold (PWT) using von Frey filaments of varying forces (**B**) and the duration of the inflamed hind paw contact with the hot plate (**C**). Results are presented as mean ± SD (*n* = 7 for each group). Statistical significance was determined using one-way ANOVA with post hoc Tukey’s test, denoted as * *p* < 0.05, ** *p* < 0.01, *** *p* < 0.001 for significant differences compared to control; # *p* < 0.05 for comparisons between experimental groups; and ns—not significant.

**Table 1 marinedrugs-22-00382-t001:** Interaction scores for 5 models of interaction between peptides wtUgr9a-1, S16K, and A23K and the ASIC1a channel with the highest negative ratio of interaction score to root mean square deviation (RMSD).

	1	2	3	4	5	Mean ± S.E.M.
wtUgr9a-1	−20.092	−14.285	−11.330	−13.981	−6.628	−13.263 ± 3.427
S16K	−24.197	−17.937	−17.211	−14.510	−13.917	−17.554 ± 2.810
A23K	−21.513	−20.793	−18.842	−18.171	−13.359	−18.536 ± 2.216

## Data Availability

All data are contained within the article.

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
