# Peer review of "Mutagenesis of the Peptide Inhibitor of ASIC3 Channel Introduces Binding to Thumb Domain of ASIC1a but Reduces Analgesic Activity"

_marinedrugs, 2024, doi:10.3390/md22090382_

Round 1

Reviewer 1 Report

Comments and Suggestions for Authors

Timur et al. are reporting an interesting study that combines computational modeling and wet-lab validations to assess Peptide inhibitors of ASIC3 channel. The following are specific questions and comments the reviewer would like to share. 

1. For Figure 1, residues shown as sticks in Mamb-1 and PcTx1 are not labelled. It is suggested to label residues as authors’ have done in Ugr9-1.

2. In Figure 2B, K16 is abnormal. The side chain is twisted with unreasonable bond length and angle. It must be corrected.

3. Also, for Figure 2, it is recommended to hide non-polar hydrogen atoms to make the plot neat and easy to read.

4. Using molecular docking to assess protein-protein interactions can result in quantitative values. Interaction energies and docking scores can be reported to guide the discussion on binding affinities. 

5. Molecular dynamics simulations can be conducted to assess the contacting patterns in a dynamic perspective. It will also be a decent way to assess whether observed contacts can be well-maintained .  

6. Figure 4F gives the reviewer a hard time to understand. What do wt, D355A, and E357A groups compare to? Usually, if wt group is not used as a control, then a control group should be plotted as the first bar.

Overall, a revision would be recommended. 

Comments on the Quality of English Language

Minor editing of English language required.

Author Response

We thank the Reviewer for the valuable comments. Below are the answers to the questions raised.

Point 1: For Figure 1, residues shown as sticks in Mamb-1 and PcTx1 are not labelled. It is suggested to label residues as authors’ have done in Ugr9-1.

Response 1: Figure 1 was changed accordingly. The indicated residues in Mamb-1 and PcTx1 were labelled.

Point 2: In Figure 2B, K16 is abnormal. The side chain is twisted with unreasonable bond length and angle. It must be corrected.

Response 2: The correction has been made.

Point 3: Also, for Figure 2, it is recommended to hide non-polar hydrogen atoms to make the plot neat and easy to read.

Response 3: The correction has been made.

Point 4: Using molecular docking to assess protein-protein interactions can result in quantitative values. Interaction energies and docking scores can be reported to guide the discussion on binding affinities.

Response 4: As quantitative results, the text presents the values of the parameter I_sc (interaction score), which is a combination of different energy interactions. Because the binding patterns of wtUgr, D4K and V20K to the channel locate the peptides in spatially inaccessible regions, individual energy parameters do not provide a clear picture. At the same time, we have shown that the I_sc value on average correlates with the electrophysiological results of the action of peptides on channels.

Point 5: Molecular dynamics simulations can be conducted to assess the contacting patterns in a dynamic perspective. It will also be a decent way to assess whether observed contacts can be well-maintained.

Response 5: Although molecular dynamics simulations are a good tool for assessing binding in dynamics, this method was not used in this study. Since we were primarily interested in the possibility of ligand binding to the channel and identifying the amino acid residues involved in this, we focused on creating static models. Some of them oriented peptides to spatially inaccessible regions of the channel (intracellular part, transmembrane domain). We found residues D355 and E357 of the channel important for the S16K and A23K peptides binding and confirmed it by site directed mutagenesis. In this case, molecular dynamics most probably cannot provide additional information, since the interaction of K26-D355 is crucial, as was shown in electrophisiological experiments.

Point 6: Figure 4F gives the reviewer a hard time to understand. What do wt, D355A, and E357A groups compare to? Usually, if wt group is not used as a control, then a control group should be plotted as the first bar.

Response 6: We corrected figure 4F (5F in the last version of the manuscript). We added additional information in the figure legend- “A value of 1.0, represented by a dotted line, corresponds to control measurements at pH 5.5 for each form of ASIC1a”. Currents were normalized to control current to compare the effect of peptide on each form of ASIC1a (wt, D355A, and E357A).

Reviewer 2 Report

Comments and Suggestions for Authors

The authors examined the effects of the peptide A23K, a mutated peptide from original Ugr9-1 (selective ASIC3 inhibitor), on homomeric ASIC1a and ASIC3 currents, which displays a significant inhibition on both currents. As previous reported, ASIC1a and ASIC3 might participate the pain modulation in peripheral nervous system (PNS). In this study, the authors found that A23K produced a significantly less pronounced analgesic effect as compared to the original wide-type peptide Ugr9-1.

The pain modulation by ASICs is very complicated, we do not know whether heteromeric channels such ASIC1a/3 also involved in pain modulation, therefore, it is also critical to see whether A23K itself affects the current of the heteromeric ASIC1a/3.

Major consideration:

1, it is good to know the effects of A23K on heteromeric channels of ASIC1a/3.

2, it is better to see whether A23K has any effects on pain modulation on ASIC2 knockout mice.

Author Response

Point 1: The pain modulation by ASICs is very complicated, we do not know whether heteromeric channels such ASIC1a/3 also involved in pain modulation, therefore, it is also critical to see whether A23K itself affects the current of the heteromeric ASIC1a/3.

It is good to know the effects of A23K on heteromeric channels of ASIC1a/3.

Response 1: We checked the effects of A23K on heteromeric ASIC1a/3 channels. As in the case of homomeric channels, the peptide had an inhibitory effect on heteromeric channels. Please see the data in the revised version (p.6, new Figure 4).

Point 2: It is better to see whether A23K has any effects on pain modulation on ASIC2 knockout mice.

Response 2: Unfortunately, we are unable to perform experiments on knockout mice to robustly validate the hypothesis.

Round 2

Reviewer 1 Report

Comments and Suggestions for Authors

Authors have revised the manuscript accordingly. No additional comments to share. 

Comments on the Quality of English Language

N/A

Author Response

Thank you for reviewing our manuscript!